Seed dormancy types and germination characteristics of six plants in the dry-warm valley of Jinshajiang River, SW China

Dong Lijuan 1 2
Geng Boyang 3
Xu Yuting 3
Yu Wei 3
Yang Li-E yanglie@ynnu.edu.cn 1 2
Peng Deli pengdeli@ynnu.edu.cn 3 4
1 Faculty of Geography, Yunnan Normal University , Kunming , Yunnan , China
2 Yunnan Provincial Key Laboratory of Plateau Geographic Process and Environmental Change, Yunnan Normal University , Kunming , Yunnan , China
3 School of Life Science, Yunnan Normal University , Kunming , Yunnan , China
4 Engineering Research Center of Sustainable Development and Utilization of Biomass Energy, Ministry of Education, Yunnan Normal University , Kunming , Yunnan , China
Gottdenker Nicole
Electronic publication date: 2025 Jun 18
Publication date: 2025
Volume: 13
Electronic Location ID: e19559
Received 2025 Feb 21; Accepted 2025 May 12
Copyright: ©2025 Dong et al.
Copyright year: 2025
Copyright holder: Dong et al.
License: This is an open access article distributed under the terms of the Creative Commons Attribution License, which permits unrestricted use, distribution, reproduction and adaptation in any medium and for any purpose provided that it is properly attributed. For attribution, the original author(s), title, publication source (PeerJ) and either DOI or URL of the article must be cited.
License URL: https://creativecommons.org/licenses/by/4.0/

Keywords: Cardinal temperatures, Dry valley, Dry after-ripening, Non-deep physiological dormancy, Light requirement

Funding: Yunnan Fundamental Research Projects 202501AT070007 This study was supported by Yunnan Fundamental Research Projects (grant NO. 202501AT070007). The funders had no role in study design, data collection and analysis, decision to publish, or preparation of the manuscript.

==============================
Seed dormancy and the requirements for germination following dormancy release are critical factors influencing the success of seedling establishment. This study examined six plant species from the dry-warm valley region of the Jinshajiang River in southwestern China, investigating their seed dormancy types and germination characteristics. Initially, germination tests were conducted using freshly matured seeds at alternating temperatures of 25/15 and 15/5 °C under light conditions. Subsequently, after dry after-ripening (DAR), germination was retested. Additionally, dried seeds were incubated under a range of constant temperatures (5–37 °C) under light conditions. The effects of darkness and GA3 on seed germination were evaluated at alternating temperatures of 25/15 and 15/5 °C. Cardinal temperatures and thermal time requirements for 50% final germination (θ50) were determined. The increase in final germination following seed coat scarification indicated that Sophora davidii seeds exhibited physical dormancy at dispersal. Treatment with DAR and/or GA3 effectively alleviated dormancy in the other five species (Osteomeles schwerinae, Excoecaria acerifolia, Leonurus japonicus, Incarvillea arguta, Berberis concolor), particularly at the cooler temperature regime of 15/5 °C, suggesting that these species possess non-deep physiological dormancy. Once dormancy is alleviated, seeds of all six plant species can germinate across a broad temperature spectrum, and the temperature window (Tb-Tc) for germination is much wider than the actual germination range. Alternating temperatures did not significantly enhance germination rates compared to constant temperatures, except for L. japonicus. Seeds of L. japonicus exhibited a strict light requirement for germination at alternating temperatures of 25/15 and 15/5 °C, whereas the other five plant species germinated effectively in darkness at the warmer alternating temperature of 25/15 °C. Thus, our hypothesis that dormancy and germination traits restrict germination to the summer (rainy season) is supported. This ensures that seedlings can establish themselves once soil moisture and temperature conditions become favorable. This research offers a valuable scientific reference for vegetation restoration efforts in dry-warm valley regions.

Introduction

Seedling establishment represents the most critical phase in a plant’s early life history (Baskin & Baskin, 2014; Chen et al., 2020). Seed dormancy ensures that germination occurs during optimal periods for seedling establishment and successful completion of the life cycle (Finch-Savage & Leubner-Metzger, 2006). This mechanism serves as an important adaptation strategy for numerous plant species, enabling them to synchronize seed germination and seedling establishment with environmental conditions (Baskin & Baskin, 2014; Lamont & Pausas, 2023). Therefore, understanding the mechanisms of dormancy release and germination responses to environmental factors is essential not only for elucidating plant ecological adaptation in specialized habitats but also for formulating effective strategies for vegetation restoration (Lai et al., 2016; Wang et al., 2024).

The requirements for dormancy release can vary depending on the type of seed dormancy (Lamont & Pausas, 2023; Wang et al., 2024; Yang et al., 2020). Baskin & Baskin (2014), Baskin & Baskin (2004), Baskin & Baskin (2021) proposed a comprehensive classification system that includes five classes of seed dormancy: physical dormancy (PY), physiological dormancy (PD), morphological dormancy (MD), morphophysiological dormancy (MPD), combinational dormancy (PY + PD). Furthermore, several dormancy classes encompass multiple lower hierarchical categories, such as subclasses and levels, leading to a diverse array of dormancy types (Baskin & Baskin, 2021). The effectiveness of dormancy release is contingent upon the specific characteristics of the dormancy class and the means employed. For instance, PD can be alleviated by plant growth regulators like gibberellic acid (GA3) or temperature treatments (warm, cold, warm + cold), depending on the species. In some species with non-deep PD, dormancy can also be released during dry storage, a process known as after-ripening (Baskin & Baskin, 2020).

Once seed dormancy is released, germination is influenced by various environmental factors including temperature, light, and soil moisture (Baskin & Baskin, 2014). Temperature is a critical factor regulating seed germination, affecting both the germination rate and final germination percentage. It also exhibits a strong correlation with ambient environmental temperatures (Arana et al., 2016; Mattana et al., 2023). In particular, the effect of temperature on seed germination is characterized by specific threshold ranges, including base temperature (Tb), optimum temperature (To), and maximum or ceiling temperature (Tc) (Cristaudo et al., 2019). Germination is inhibited at temperatures below Tb and above Tc. The temperature range within which seeds germinate at the highest rate is defined as To. The thermal-time approach can be used to estimate the threshold temperature ranges for germination in a given plant species (Rezaei-Manesh et al., 2023). In seasonal climates, temperature serves as a reliable indicator of the time of year and is therefore closely linked to the timing of germination (Porceddu et al., 2013). Besides temperature, light can also influence germination in the field; however, seed responsiveness to light varies among species. For instance, some species require either light or darkness for germination, while others exhibit equal germination percentages in both light and dark conditions (Baskin & Baskin, 2014). A light requirement may inhibit germination in deep soil, and when disturbances occur that unearth the seeds, they initiate germination in locations and at times that favor seedling establishment (Peng et al., 2021; Wang et al., 2017). Furthermore, the interaction between light and temperature can alter seed sensitivity to both factors (Luna et al., 2022). Consequently, the requirements for temperature and light during germination are generally linked to the range of environmental conditions to which a given species is adapted, ensuring that germination timing aligns with favorable conditions for subsequent seedling establishment, particularly in extremely harsh ecosystems (Peng et al., 2021; Wang et al., 2017; Hu et al., 2015).

The dry-warm valley (DWV) of the Jinshajiang River, primarily situated in the upper reaches of the Yangtze River in China, represents the lowest elevational zone within mountain ecosystems. This region exhibits a markedly drier and warmer climate compared to its surroundings and to other areas at similar latitudes (Zong et al., 2022; Zhang, 1992). Under such harsh environmental conditions, the natural vegetation is predominantly composed of herbaceous plants, interspersed with sparse trees (Zhang, 1992). Owing to its highly heterogeneous environment, the DWV region has emerged as one of the most significant conservation areas and critical biodiversity hotspots in southwestern China, characterized by a high concentration of endemic plant species (Tang et al., 2020). However, the regeneration and restoration of natural vegetation occur at an exceptionally slow pace. Plants possess adaptations that enhance seedling survival and growth across diverse environments (Baskin & Baskin, 2014). The success of seedling establishment is critically dependent on seed germination, as this process dictates the timing and location of seedling establishment. Yet, our understanding of the germination characteristics and seed dormancy types of plants in the DWV region remains limited.

We hypothesized that seeds possess dormancy mechanisms and/or specific germination requirements, enabling them to germinate exclusively during the rainy season (summer) in the DWV region. To test this hypothesis, we selected six plant species from a natural community in the DWV region of the Jinshajiang River and investigated their seed dormancy types and germination characteristics. Specifically, we addressed the following questions: (1) What are the dormancy types for each species, and how is dormancy released? (2) How do different temperatures influence dormancy and germination, and what are the cardinal temperatures and thermal time requirements for germination? (3) How do light conditions affect germination? Based on laboratory measurements of seed dormancy and germination responses to temperature, GA3, and light, we aim to provide valuable data to support vegetation restoration efforts in the DWV region of the Jinshajiang River.

Materials & Methods

Study species and seed collection

The town of Ben-zi-lan is located along the Jinshajiang River in the DWV region of Deqin County, Yunnan Province, southwestern China. The rainy season in this region is short, with most rainfall occurring between July and September, resulting in generally low annual precipitation (Jin, 2002). Other seasons are marked by a significant lack of precipitation, leading to extremely arid conditions. Temperature extremes in this region range from a maximum of 37 °C to a minimum of −4 °C (Luo & Zhou, 2008). This study examines six plant species native to this region: Sophora davidii (Fabaceae), Osteomeles schwerinae (Rosaceae), Excoecaria acerifolia (Euphorbiaceae), Leonurus japonicas (Lamiaceae), Incarvillea arguta (Bignoniaceae), Berberis concolor (Berberidaceae). Mature fruits of all study species were collected in early October 2014 from multiple plants growing in the DWV region near Ben-zi-lan town, located in the upper Jinshajiang River (28°24′N, 99°31′E, 2,650–2,750 m a.s.l.). The average annual temperature in Ben-zi-lan town is 18.9 °C, with the hottest month’s average temperature at 25.7 °C and the coldest month’s average temperature at 10.4 °C. The annual precipitation is 134 mm, with the wettest month’s precipitation at 79 mm and the driest month’s precipitation at 19 mm (Yu, 2018). In the laboratory, non-seed structures and empty seeds were manually removed. The cleaned seeds were divided into two groups: one group was used immediately for the experiment as the control group (fresh), while the other group was placed in paper bags and air-dried (DAR) at room temperature (13–22 °C; 25–55% relative humidity) for six months. Three groups of 100 air-dried mature seeds were randomly selected and weighed using a ten-thousandth analytical balance. The average seed weight was then calculated.

Seed imbibition

To assess the permeability of seeds to water, four replicates of 25 whole seeds and/or scarified seeds of each species were placed on moist filter papers at room temperature (approximately 20 °C). The seeds of Sophora davidii were divided into two groups. In the first group, the seed coats were softened by soaking in water and then gently nicked with forceps to break the outer seed coat. The second group received no treatment and was used directly in the water absorption experiment. Seeds were weighed at time 0 (W1) and subsequently at 2, 4, 6, 8, 10, 12, 24, 36, 48, 60, and 72 h. At each weighing interval, seeds were removed from the moist paper, blotted dry, reweighed (W2), and then returned to the moist filter paper. Water uptake by the seeds was calculated using the formula: Water absorption (%) = [(W2-W1)/W1] × 100. A significant increase in seed mass indicates permeable seed coats, whereas minimal or no increase in mass suggests water-impermeable coats, indicative of physical dormancy (PY) (Baskin & Baskin, 2014).

Germination tests of fresh seeds and DAR seeds

Experiments on fresh seeds of all species were initiated one week after collection, using 1% water agar as the substrate in 90 mm plastic Petri dishes. Two alternating temperature regimes were employed: 15/5 and 25/15 °C (12/12 h). The photoperiod was set to 12 h light at 22 µmol m−2 s−1 from cool white fluorescent light (400–700 nm) during the higher temperature phase, followed by 12 h of darkness (hereafter referred to as “in the light”). After approximately six months, DAR seeds were transferred to germination conditions (15/5 and 25/15 °C) starting in April 2015. To assess the presence and level of PD post-DAR, seeds were incubated on 1% water agar supplemented with 100 mg L−1 of GA3 at the aforementioned temperatures. To evaluate the effect of darkness on germination, DAR seeds were tested under continuous darkness and light conditions at both 15/5 and 25/15 °C. Continuous darkness was achieved by wrapping Petri dishes with two layers of aluminum foil. To determine the optimal temperature for germination, DAR seeds from all species were incubated at eight constant temperatures (5, 10, 15, 20, 25, 30, 35, and 37 °C) under light conditions for a period of 42 days.

Due to the limited availability of well-developed seeds, three replicates of 20–25 seeds each were used under each condition. The Petri dishes were placed in transparent plastic bags to prevent desiccation. Seeds incubated under light conditions were monitored daily, and germinated seeds were removed immediately. In contrast, seeds incubated in darkness were counted only once at the end of the experiment to prevent any exposure to light (42 days). Germination was defined by visible radicle emergence. Germination ceased after 3–5 weeks, and all experiments were terminated after 6 weeks. The viability of ungerminated seeds was assessed using a cut-test, where seeds with plump, firm, and white embryos were deemed viable.

Statistical analysis

Final germination percentage (GP) and mean germination time (MGT) were calculated using the following formulas: GP (%) = (∑Gi/N) ×100, MGT (days) = ∑(i×Gi)/∑Gi, where i represents the day of germination relative to the sowing date, Gi is the number of seeds germinated on day i, and N is the total number of filled seeds. Statistical analysis was conducted using SPSS version 26.0 (IBM Corporation, Chicago, IL, USA). The final germination percentage was reported as the mean ± standard error (SE) based on three replicates. Independent samples t-tests were conducted to evaluate: (1) whether DAR and GA3 significantly increased germination percentage (GP) compared to fresh seeds; (2) whether alternating temperatures significantly accelerated germination (GP) relative to constant temperatures (specifically, 15/5 vs. 10 °C and 25/15 vs. 20 °C); (3) whether dark conditions significantly inhibited germination more than light conditions. One-way ANOVA was employed to compare the GP and MGT of non-dormant seeds under constant temperatures in light conditions.

A thermal-time model was utilized to estimate cardinal temperatures following the methodology outlined by Hardegree (2006). Germination time courses from all three replicates at each temperature were pooled and fitted using the Weibull function in Origin 2022 (OriginLab Corporation, Northampton, MA, USA; http://www.originlab.com) as described by Peng et al. (2021). The time to reach 50% final germination (t 50) was determined by fitting cumulative germination progress curves. The germination rate (GR, defined as 1/t 50) was plotted against temperature and regressed using a linear model. The base temperature (Tb), at which the germination rate is zero, and the cardinal temperature (Tc), the maximum temperature for germination, were estimated from this regression. The optimum temperature (To) for germination was calculated as the intersection of the sub-optimal and supra-optimal temperature-response functions. The slope of the linear regression line represented the reciprocal of the thermal-time requirement (θ50) at both suboptimal and supra-optimal temperatures (Geng & Peng, 2022). Statistical analyses were conducted using the Statistical Package for Social Sciences (SPSS) version 26.0 (IBM Corp., Armonk, NY, USA). Data are presented as mean ± standard error (SE). All figures were generated using Origin 2022.

Results

Weight of the 100 seeds

The 100 seed weights were (n = 10): 1.653 ± 0.056 g (Sophora davidii), 0.692 ± 0.014 g (Osteomeles schwerinae), 0.226 ± 0.008 g (Excoecaria acerifolia), 0.123 ± 0.002 g (Incarvillea arguta), 0.048 ± 0.001 g (Leonurus japonicus), 1.04 ± 0.034 g (Berberis concolor).

Water imbibition test

The seeds of five species, excluding S. davidii, imbibed rapidly, with their mass increasing by 20–140% within the first 12 h, after which there was minimal further increase in mass (Fig. 1). In contrast, manually scarified seeds of S. davidii reached over 90% of their initial mass, while intact seeds showed no significant mass increase (Fig. 1). Therefore, apart from S. davidii, the seeds of the other five species exhibited water-permeable seed coats.

Figure 1 Imbibition curves of intact (black line) or manually scarified (red line) seeds and morphological structure of seeds of six study species.

Water uptake data are means ± SE (n = 4). The dashed line indicates the outline of the embryo; for exalbuminous seeds, the embryo is not drawn separately with a dashed line.

Effect of DAR, GA3 and light/dark on germination

Fresh seeds of all species incubated in light germinated 30–80% under warmer temperatures (25/15 °C). However, GP were significantly lower at cooler temperatures (15/5 °C), with only approximately 10% observed in S. davidii and L. japonicus, and no germination in O. schwerinae, E. acerifolia, I. arguta, and B. concolor (Fig. 2). After DAR for six months, GP increased at both temperature regimes, with significant interspecific differences noted (Fig. 2). In intact seeds of S. davidii, the effect of DAR was not significant, whereas it was positive and statistically significant in the other five species at both temperatures (except at 25/15 °C for I. arguta) (Fig. 2). GA3 did not enhance germination of DAR seeds in any species at 25/15 °C, but significantly increased germination in DAR seeds of E. acerifolia, L. japonicus, and I. arguta at 15/5 °C (Fig. 2). Germination of DAR seeds was generally higher under light conditions compared to darkness, irrespective of species; however, light did not have a positive effect on E. acerifolia at either 25/15 °C or 15/5 °C, nor on scarified seeds of S. davidii at 15/5 °C (Fig. 3).

Figure 2 Final germination percentages (n = 3, means ± SE) at two alternating temperatures for fresh, dry after-ripened (DAR) and DAR + GA3 seeds of six species.

All germination percentages shown in this figure were obtained under light conditions. 0, no germination at this treatment. Bars with different uppercase letters indicate significant differences (P < 0.05) in germination percentage between fresh and DAR seeds, and different lowercase letters indicate significant differences (P < 0.05) between DAR and DAR + GA3 seeds at each temperature.

Figure 3 Effects of light condition on germination percentage (n = 3, means ± SE) for DAR seeds of six species at two alternating temperatures.

The seeds of Sophora davidii in this figure were subjected to seed coat scarification. 0, no seeds germinated in this treatment. * P < 0.05, ** P < 0.01, and *** P < 0.001, ns, not significant.

Germination behaviors at different temperatures

The germination of all species following DAR was significantly influenced by temperature (5−37 °C, Fig. 4). Seeds subjected to DAR demonstrated high GP and rapid germination (low MGT) at temperatures of 15, 20, and 25 °C. However, GP and the speed of germination markedly decreased at temperatures below 10 °C or above 30 °C, with no germination observed at the extremes of 5 and 37 °C (Fig. 4). Incubation at alternating temperatures of 25/15 °C did not significantly affect GP compared to the constant temperature of 20 °C, with the exception of scarified seeds of S. davidii. Incubation at 15/5 °C resulted in higher GP for S. davidii and L. japonicus compared to the constant temperature of 10 °C, while it led to decreased GP and/or slower germination speed in E. acerifolia and I. arguta (Fig. 5).

Figure 4 Effects of temperature on germination percentage and mean germination time (MGT) (n = 3, means ± SE) for DAR seeds of six species at light condition.

The seeds of Sophora davidii in this figure were subjected to seed coat scarification. ×, no test in this treatment; 0, no germination at this temperature. Bars with different uppercase letters indicate significant differences (P < 0.05) for germination percentage, and scatters with different lowercase letters indicate significant differences (P < 0.05) for MGT at different temperatures.

Figure 5 Effect of alternating and corresponding constant temperatures (LT, low temperature; 15/5 vs. 10 °C; HT, high temperature; 25/15 vs. 20 °C) on final germination percentage (n = 3, means ± SE).

All germination percentages shown in this figure were obtained under light conditions. 0, no seeds germinated in this treatment. ** P < 0.01, ns, not significant.

Thermal requirement for germination

The germination of seeds from all species in response to temperature was accurately described by the thermal-time model across both sub-optimal and supra-optimal temperature ranges (Fig. 6). The results demonstrated significant variations in the temperature thresholds required for seed germination among different species (Table 1). Tb ranged from 4.7 to 6.7 °C, and Tc ranged from 35.6 to 38.5 °C across all species. The To of L. japonicus was the highest at 34.08 °C, while the other species exhibited a narrower range of 20.75 to 28.24 °C. B. concolor demonstrated a broader To range, spanning from 15 to 25 °C. At suboptimal temperatures, thermal times (θ50) differed significantly among the species examined. O. schwerinae exhibited the highest recorded value of 190.11 °C ⋅ d, whereas the other species displayed relatively lower values, ranging from 33.4 to 57.47 °C ⋅ d (Table 1).

Figure 6 Curve fitting of linear model to germination rate (1/t50) versus temperature at constant temperatures (5–37 °C) for DAR seeds in six species.

Germination rates (n = 3, means ± SE) calculated on the basis of the reciprocal of the times to reach 50% final germination. Points correspond to the actual data and red solid lines indicate the fitted lines from the linear regressions.

Table 1 Estimation of temperature threshold value with a linear regression of seed germination rate 1/t50 as a function of temperature in six species.

Species	Tb (°C)	To (°C)	Tc (°C)	θ50 (°C ⋅ d)	Regression equation	R 2	
Sophora davidii	4.71	26.27	38.02	33.4	y = 0.0299x − 0.141	0.8	
y =  − 0.0549x + 2.091	0.97	
Osteomeles schwerinae	4.94	21.55	35.65	190.11	y = 0.0053x − 0.026	0.99	
y =  − 0.0062x + 0.221	0.95	
Excoecaria acerifolia	6.73	28.24	37.49	52.52	y = 0.0190x − 0.128	0.94	
y =  − 0.0443x + 1.659	0.97	
Leonurus japonicus	5.56	34.08	37	57.47	y = 0.0174x − 0.097	0.99	
y =  − 0.1705x + 6.309	1	
Incarvillea arguta	4.9	26.06	38.53	41.49	y = 0.0241x − 0.118	0.95	
y =  − 0.0409x + 1.577	0.78	
Berberis concolor	5.27	20.75	35.74	109.89	y = 0.0091x − 0.048	0.99	
y =  − 0.0094x + 0.336	0.94	
Notes.

θ50 is the thermal-time requirement for germination at suboptimal temperatures.

Discussion

This study investigated the seed dormancy types and germination characteristics of six plant species in the DWV region of the Jinshajiang River, southwestern China. In our experiments, we systematically employed a variety of methods to release dormancy and promote germination. Our results demonstrated that all six species exhibited some form of seed dormancy; five species displayed PD, while one species exhibited PY. The application of DAR and GA3 proved to be the most effective treatments for breaking seed dormancy. Once dormancy is released, seeds can germinate across a broad temperature spectrum. However, the effect of light on germination is both temperature-dependent and species-specific. These dormancy mechanisms, along with the specific light and temperature requirements for germination, may delay seed germination until the rainy season in the field, when both temperature and soil moisture are optimal. This represents an advantageous ecological adaptation to adverse valley environments.

Type of dormancy

In this study, the six species exhibiting dormancy can be categorized into two types, PY and PD, based on the classification system proposed by Baskin & Baskin (2021). For S. davidii, manually scarified seeds demonstrated significantly higher water imbibition compared to intact seeds (Fig. 1), indicating that this species possesses PY. Notably, neither six months of DAR nor GA3 treatments were effective in breaking dormancy (Fig. 2). The embryos of freshly matured seeds in O. schwerinae, E. acerifolia, L. japonicus, I. arguta, and B. concolor were fully developed (Fig. 1), and intact non-treated seeds exhibited water imbibition levels ranging from 30% to 80% (Fig. 1). Following dormancy-breaking treatments, the GP increased significantly (Fig. 2). These imbibition and germination experiments clearly demonstrated that fresh seeds of the five species, excluding S. davidii, lack physical dormancy (PY) and exhibit only physiological dormancy (PD). Three levels of PD were identified: non-deep, intermediate, and deep (Baskin & Baskin, 2021). Non-deep PD is the most prevalent form and can be alleviated by brief periods of chilling stratification, DAR (dry storage), or GA3 treatment (Baskin & Baskin, 2014). In this study, DAR effectively broke the dormancy, confirming that these species possess non-deep PD. Mature seeds with primary dormancy (PY or PD) are dispersed from maternal plants during the dry season, preventing immediate germination and forming a transient seed bank in the soil. Primary dormancy is alleviated by the temperature and moisture conditions in the soil during the dry (winter) season, while the high temperatures of the rainy (summer) season stimulate germination. This mechanism allows plants to fully exploit the distinct rainy and dry seasons.

Thermal requirement for germination after dormancy release

For successful seedling establishment, seeds must germinate under optimal environmental conditions. The timing of germination is determined by two key factors: seeds must first break dormancy, and subsequently, the specific germination requirements of nondormant seeds must be met (Soltani, Baskin & Gonzalez-Andujar, 2022). These requirements encompass temperature, light, moisture, and other environmental cues present in the habitat (Baskin & Baskin, 2014; Ma, Erickson & Merritt, 2018), with temperature and light being particularly critical (Baskin & Baskin, 2014; Luna et al., 2022; Ma, Erickson & Merritt, 2018). Once dormancy is alleviated, seeds will initiate germination within a specific temperature range. This temperature range, defined by the cardinal temperatures, plays a pivotal role in shaping seed germination characteristics (Soltani, Baskin & Gonzalez-Andujar, 2022). In this study, seeds of all tested plant species were capable of germinating within a broad temperature range of 10–30 (35) °C (Fig. 4). Furthermore, alternating temperatures did not significantly enhance GP compared to constant temperatures, with the exception of L. japonicus (Fig. 5). Values of thermal time and cardinal temperatures provide a context for adaptive strategies optimizing the efficiency of germination in relation to temperature (Maleki et al., 2024). According to the thermal-time model, the Tb for germination ranged from 4.71 to 6.73 °C, and the Tc varied between 35.6 to 38.5 °C (Table 1). The θ50 requirements for germination were also relatively low (33.4–190.11 °C ⋅ d) (Table 1; Fig. 6). So, the temperature window (Tb − Tc) for germination is much wider than the actual germination range, which allows seeds to germinate at high or at low temperatures, resulting in many seedling cohorts that emerged during the rainy season when soil moisture is non-limiting. Moreover, the Tb for germination is lower than the annual temperature range of 6–16 °C in the DWV region (Zhang, 1992). Consequently, the lower Tb and θ50 enable seeds to accumulate sufficient heat units to meet their thermal requirements for germination when ambient temperatures are relatively high. This adaptation is a result of long-term evolutionary responses to the unique climatic conditions of the DWV region.

Light requirement for germination after dormancy release

The light responses of seeds play a crucial role in ensuring germination occurs in suitable locations that are conducive to seedling establishment (Peng et al., 2021; Wang et al., 2021). In the DWV region, the soil surface is often exposed to high temperatures, elevated evaporation, and low moisture levels, even during the rainy season (Jin, 2002), which can inhibit both seed germination and seedling establishment. Burying seeds can significantly mitigate these environmental stresses by reducing exposure to high temperatures and drought conditions, although it also decreases light availability. Consequently, dark germination would be an adaptive strategy for species inhabiting the DWV region. In this study, the seeds of all species except L. japonicus germinated well in darkness, particularly at the warmer temperatures typical of the rainy season (25/15 °C), suggesting that burial in soil does not completely inhibit germination (Fig. 3). While burial in soil or the ground layer may create favorable conditions for seed germination, it might hinder seedling emergence to the surface. In this study, the seed mass of all six species exceeded 0.1 mg, classifying them as “large seeds” according to Grime et al. (1981). Large seeds typically do not require light for germination. Furthermore, larger seeds exhibit greater biomass growth, enhancing seedling emergence and survival under low-light conditions (Grime et al., 1981; Ma et al., 2019). Consequently, the light/dark responses of seed germination confer a survival advantage under unfavorable conditions in the DWV region.

A light requirement for germination is an efficient mechanism for detecting gaps and allows seeds to respond to disturbances (Luna et al., 2022). We observed that L. japonicus seeds exhibited a more stringent light requirement for germination compared to seeds of the other five species, irrespective of temperature regime (Fig. 3). This phenomenon can be partly attributed to the species’ distribution in disturbed habitats. Under natural conditions, we found that the highest seedling emergence percentage of L. japonicus occurred in microhabitats characterized by vegetation or soil disturbance. Consequently, both light availability and alternating temperatures (15/5 vs. 10 °C, Fig. 5) trigger seed germination of L. japonicus following disturbance events. Although L. japonicus is widely distributed, it maintains small population sizes in disturbed areas, whereas S. davidii, O. schwerinae, and E. acerifolia are dominant species in the local flora of the Jinshajiang River region. This highlights L. japonicus’s strong adaptation to germination under varying light conditions in the DWV region.

Seed responses to light can significantly influence the timing of germination from the soil seed bank (Wang et al., 2017). A light requirement may inhibit germination in deep soil layers, thereby promoting the formation of persistent soil seed banks (Peng et al., 2018; Peng et al., 2021). In this study, darkness inhibited seed germination across all species at low alternating temperatures (15/5 °C) during the dry season (Fig. 3), preventing some seeds from germinating when buried deeply or shallowly in the soil or ground layer. These germination characteristics suggest that plants possess the ability to form persistent soil seed banks (Peng et al., 2021). Seeds can remain dormant in the soil seed bank if water availability is insufficient for germination or if the release of PY and PD remains incomplete, delaying germination until soil moisture and temperature conditions become favorable in subsequent years.

Conclusions

In summary, fresh seeds of all six tested species exhibited dormancy at dispersal. Notably, S. davidii displayed PY, while the other five species demonstrated non-deep PD. These findings suggest that non-deep PD may be prevalent in the DWV region of the Jinshajiang River. The scarification method employed in this study was sufficient to break PY, and DAR proved to be an effective approach to alleviate PD. After dormancy release, seeds exhibit a broad temperature range (Tb − Tc) for germination during the rainy season. However, at cooler temperatures (15/5 °C), seeds require more light for successful germination compared to warmer temperatures (25/15 °C). These dormancy-breaking and germination requirements ensure that seeds do not germinate immediately after dispersal during the dry season (winter). Instead, they promote germination in the subsequent rainy season (summer) or contribute to the establishment of persistent soil seed banks. This supports our hypothesis that seeds have specific dormancy mechanisms and/or germination requirements, allowing them to germinate exclusively during the rainy season (summer). This ensures that seedlings establish only when soil moisture and temperature conditions are optimal. For vegetation restoration, sowing DAR or scarified seeds in the early rainy season would increase the likelihood of germination. The precipitation and temperature conditions during this period are optimal for seedling survival.

Supplemental Information

Supplemental Information 1 Seed germination data

Supplemental Information 2 Seed imbibition experiment data

The authors are grateful to Qiangbang Gong, Na Chen and Fan Wang for their assistance in laboratory work.

Additional Information and Declarations

Competing Interests

Author Contributions

Data Availability

The authors declare there are no competing interests.

Lijuan Dong performed the experiments, analyzed the data, prepared figures and/or tables, authored or reviewed drafts of the article, and approved the final draft.

Boyang Geng analyzed the data, prepared figures and/or tables, and approved the final draft.

Yuting Xu performed the experiments, prepared figures and/or tables, and approved the final draft.

Wei Yu performed the experiments, prepared figures and/or tables, and approved the final draft.

Li-E. Yang conceived and designed the experiments, authored or reviewed drafts of the article, and approved the final draft.

Deli Peng conceived and designed the experiments, performed the experiments, authored or reviewed drafts of the article, and approved the final draft.

The following information was supplied regarding data availability: Supplementary File.

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
