# Peer review of "Seed dormancy types and germination characteristics of six plants in the dry-warm valley of Jinshajiang River, SW China"

_PeerJ, doi:10.7717/peerj.19559_

## Round 0.1 · original submission · Minor Revisions

Reviewer 1 ·

Basic reporting

The authors studied the seed germination characteristics of six species of plants native to the dry warm valley area of ​​the Jinshajiang River in China.
The research background was clear, and the experimental results were obtained according to the purpose.
And it is judged that the review was sufficiently conducted.

Experimental design

The hypothesis set out in this study is persuasive and the experiment was designed to support it.

Validity of the findings

The results obtained through the experiment provide basic information that can contribute to the field of seed ecology research, and the results can also be practically utilized in restoring native habitats.

Additional comments

- line 67: five types of seed dormancy --> five classes of seed dormancy
- lines 109-112: Sentences with redundant expressions are repeated. They need to be revised.
- line 127 ~: 'DWH' --> DWV,,DWH --> Terms are mixed. This part needs to be described clearly.
- line 227: "scarified seeds of S." --> In the case of this plant, the materials and methods should clearly describe how and in which experiment the SCAR treatment was performed. If the figure data in the back is the result of SCAR-treated seeds, the materials and methods should describe the details.
- line 222: concolo --> Modify the plant species name
- line 292: '30% to 80%' --> Clearly describe where the result refers to. If it includes the result of culturing at 15 degrees, it can be 0%. - line 292: Figure 1 --> Figure 2
- line 295: 'primary dormancy' --> physical dormancy
- line 352: Leonurus --> If the genus name of a plant is repeated in the same section, write it as an abbreviation
- Figure 3: The results of Sophora show the results of SCAR-treated seeds. If so, M/M should provide a detailed explanation of this part. It would also be good to add a brief explanation to the title of the figure so that readers can easily understand the results even if they only look at the figure.
- Figure 4: The results of Sophora seem to be the results of 'DAR + SCAR' treatment. However, the title only shows DAR, which is confusing for easy understanding. It would be better to write it clearly.

·

Basic reporting

• This study aims to understand the seed dormancy types and germination characteristics of six plant species selected from the natural community.
• The English language is clear, unambiguous and professional used throughout. Only in line 140 there is the statement: “leading to overall lower annual precipitation” – no information "lower than?" this should be explained or maybe just write simple – low not lower?
• Intro & background to show context.
The introduction is well written, only there is a minor mistakes: Lines 111-113 “the natural vegetation is predominantly composed of herbaceous plants, interspersed with sparse trees, and in some areas, woodland and xerophytic thorny shrubs (Zhang, 1992)”. This phrase is repeated, it was above, see lines 109-111, so it must be deleted.
• Literature
The cited references are well selected, including the most recent publications, more than half of them are from the last 5 years.
• Structure conforms to PeerJ standards, discipline norm, or improved for clarity.
• Figures are relevant, high quality, well labelled and described.
• Raw data supplied
Thank you for providing the raw data

Experimental design

• It is original primary research within Scope of the journal.
• The authors presented the aim of the work and three questions that the results of their research are supposed to answer. However, there is no specific research hypothesis. Please add it.
• Methods were described with detail and information to replicate. However, they require small additions:
- 153 line: in how many replications was the weight of 100 seeds measured?
- 172 line: DAR – in the methods section all abbreviations should be explained.
- 178-179 line: How long were the DAR seeds incubated at eight constant temperatures and light conditions?
- 184 line: seeds incubated in darkness were counted at the end of the experiment i.e. after how many weeks?
Furthermore, I propose that the description of weather conditions for the region where the seeds were collected should be supplemented with data (temperature and precipitation) for the period of seed setting, filling and ripening of the studied species.

Validity of the findings

All underlying data have been provided; they are robust and statistically sound.
But the paper should provide the weight of 1000 seeds calculated based on the weight of 100 seeds. This value is a measure of seed size, which is commonly marked because it is important for calculating the sowing rate.
Conclusions are well stated, linked to original research questions. Moreover they are consistent with the data and the arguments presented.
Since the results are only one-year, they require repetition with seeds collected in subsequent years, in different weather conditions, because as shown in many works, both the seed mass and their germination capacity can significantly differ between years, when the seeds matured in different weather conditions (temperature, precipitation) (see e.g. Janicka et al. 2021 ).
Janicka, M.; Pawluśkiewicz, B.; Małuszyńska, E.; Gnatowski, T. 2021. Diversity of the Seed Material of Selected Plant Species of Naturally Valuable Grassland Habitats in Terms of the Prognosis of Introduction Success. Sustainability, 13, 13979. https://www.mdpi.com/2071-1050/13/24/13979

Additional comments

In my opinion this manuscript is interesting and generally well written. I have read it with interest. It contains interesting results of germination characteristics of chosen six plant species. I have to say, it was a pleasant reading, since it is detailed and sounds good in every section.
The manuscript is clear, relevant for the field of PeerJ and presented in a well-structured manner. Minor comments are included above.

·

Basic reporting

This study aimed to determine the dormancy types and germination characteristics of six plant species in the dry-warm valley of the Jinshajiang River from China. Due to the rich biodiversity of the region and the need for restoration, it is important to know the germination and dormancy characteristics of the plant species studied. In addition, the most important contribution of this study is the determination of cardinal temperatures for the germination of these species.

Experimental design

see additional comments

Validity of the findings

see additional comments

Additional comments

Additional comments:
Introduction;
Lines 108-113; there are two very similar sentences. They need to be merged (Zhang, 1992).
Lines 66-73; authors give definitions of dormancy classes by Baskin and Baskin. I think many of the readers are already know these definitions and so not needed to write detailed in here.
The ecological conditions and vegetation characteristics of the valley are given here long since the region is considered as biodiversity hotspot. I suggest to shortening it.
Materials and Methods:
Seed imbibition;
When describing the seed imbibition, it is stated that only one species was subjected to scarification. This section needs to specify how the scarification was done. I learned from the supplementary file 2, that scarification is done manually, that is, by seed coat cut.
Results:
Figure descriptions need to include whether there is a photoperiod or not, e.g. Fig 2 and Fig 5.
Germination behaviours at different temperatures;
The results of which are given in Figure 4, I understood that Sophora davidii was subjected to scarification at different temperature incubations after DAR. But this is not explained material and method.
Discussion:
line 294, ………….. five species lack primary dormancy (PY) but exhibit PD.
but also in line 295, primary dormancy (PY),

These two sentences are confusing. In the following sentences, it is not clear what the abbreviation PY stands for.

---

## Round 0.2 · accepted · Accept

All Reviewers comments have been adequately addressed, and the paper is ready for publication.

Reviewer 1 ·

Basic reporting

no comment

Experimental design

no comment

Validity of the findings

no comment

Additional comments

The authors have revised the version to reflect the reviewers' comments.

·

Basic reporting

My previous comments were taken into account.
Thank you, authors, for the updates and revisions.

Experimental design

-

Validity of the findings

-

·

Basic reporting

Suitable. No additional comments.

Experimental design

Suitable. No additional comments.

Validity of the findings

Suitable. No additional comments.

Additional comments

I thank the authors for improving the manuscript. The manuscript has been revised according to the comments.